# The Global Prevalence of Anxiety Among Medical Students: A Meta-Analysis

**DOI:** 10.3390/ijerph16152735

**Published:** 2019-07-31

**Authors:** Travis Tian-Ci Quek, Wilson Wai-San Tam, Bach X. Tran, Min Zhang, Zhisong Zhang, Cyrus Su-Hui Ho, Roger Chun-Man Ho

**Affiliations:** 1Department of Psychological Medicine, Yong Loo Lin School of Medicine, National University of Singapore, Singapore 119007, Singapore; 2Institute of Cognitive Neuroscience, Huaibei Normal University, Huaibei 235000, China; 3Alice Lee Centre for Nursing Studies, National University of Singapore, Singapore 119007, Singapore; 4Institute for Preventive Medicine and Public Health, Hanoi Medical University, Hanoi 100000, Vietnam; 5Johns Hopkins Bloomberg School of Public Health, Johns Hopkins University, Baltimore, MD 21205, USA; 6Vietnam Young Physicians’ Association, Hanoi 100000, Vietnam; 7School of Education, Huaibei Normal University, Huaibei 235000, China; 8Department of Psychological Medicine, National University Health System, Singapore 119007, Singapore; 9Institute for Health Innovation and Technology (iHealthtech), National University of Singapore, Singapore 117599, Singapore; 10Centre of Excellence in Behavioral Medicine, Nguyen Tat Thanh University (NTTU), Ho Chi Minh City 700000, Vietnam

**Keywords:** anxiety, anxious, medical students, student doctors, medical school, medical education, prevalence, meta-analysis, review

## Abstract

Anxiety, although as common and arguably as debilitating as depression, has garnered less attention, and is often undetected and undertreated in the general population. Similarly, anxiety among medical students warrants greater attention due to its significant implications. We aimed to study the global prevalence of anxiety among medical students and the associated factors predisposing medical students to anxiety. In February 2019, we carried out a systematic search for cross-sectional studies that examined the prevalence of anxiety among medical students. We computed the aggregate prevalence and pooled odds ratio (OR) using the random-effects model and used meta-regression analyses to explore the sources of heterogeneity. We pooled and analyzed data from sixty-nine studies comprising 40,348 medical students. The global prevalence rate of anxiety among medical students was 33.8% (95% Confidence Interval: 29.2–38.7%). Anxiety was most prevalent among medical students from the Middle East and Asia. Subgroup analyses by gender and year of study found no statistically significant differences in the prevalence of anxiety. About one in three medical students globally have anxiety—a prevalence rate which is substantially higher than the general population. Administrators and leaders of medical schools should take the lead in destigmatizing mental illnesses and promoting help-seeking behaviors when students are stressed and anxious. Further research is needed to identify risk factors of anxiety unique to medical students.

## 1. Introduction

Medical schools around the world aim to train and produce competent and empathetic physicians to help the sick, advance medical knowledge, and promote public health [1]. However, medical education is considered to be one of the most academically and emotionally demanding training programs out of any profession [2], and consequently, the time and emotional commitment necessary for medical students to devote to their training is extensive. Such demands and stress cause a negative effect on the students’ psychological well-being [3], and can precipitate depression and anxiety [4]. Anxiety, although as common and arguably as debilitating as depression [5], has garnered less attention and is often undetected and undertreated [6] in the general population. In addition to intense feelings of fear or panic [7], sufferers of anxiety can experience other physiological symptoms including fatigue, dizziness, headaches, nausea, abdominal pain, palpitations, shortness of breath, and urinary incontinence [8]. Anxiety can also impair goal-directed attention and concentration [9], working memory [10], and perceptual-motor function [11], all of which are important domains which enable medical students and physicians to provide safe and efficacious medical care to patients.

In the current literature, anxiety among medical students is less studied than depression. A 2014 systematic review of the prevalence of anxiety among medical students outside of North America found a large range of prevalence between 7.7% and 65.5% across 11 studies [12]. The relatively small number of studies reviewed in Hope’s study was due to the fact that it included medical students in Europe and English speaking countries outside North America. Anxiety in medical students warrants wider awareness and greater attention. It can negatively affect academic performance, dropout rates and professional development [1]. Ultimately, it could also have implications for patient care since a previous study found that anxious medical students were less empathetic and less enthusiastic when caring for patients with chronic illness [1]. Quality of patient care suffers as doctors with anxiety tend to demonstrate poor work efficacy [13].

To the best of our knowledge, there has not been a meta-analysis of the global prevalence of anxiety among medical students published in the literature. We conducted this study with the aims of bridging this gap by estimating the global prevalence of anxiety among medical students, as well as identifying epidemiological and social factors associated with anxiety in medical students in order to identify at-risk students and provide timely assistance and intervention. 

## 2. Materials and Methods

### 2.1. Review Protocol and Registration

A review protocol was not registered in any database before the initiation of this study.

### 2.2. Search Strategy

We conducted our initial literature search in February 2019. We applied a standardized search strategy (see Search strategy) to the following databases: (i) MEDLINE from 1946 to March 2019; (ii) PsycINFO from 1806 to March 2019, and (iii) Embase, from 1980 to March 2019. In addition, we screened the reference lists of identified articles to identify additional relevant articles. Searches were last updated in March 2019.

### 2.3. Inclusion Criteria

We included studies that fulfilled the following criteria: (i) assessed students who were studying medicine; (ii) analyzed the prevalence of anxiety; (iii) measured the prevalence of anxiety using standardized validated instruments and questionnaires, and (iv) published articles in English.

### 2.4. Exclusion Criteria

We excluded studies with the following characteristics: (i) did not give an aggregate prevalence of anxiety among medical students; (ii) had grouped veterinary medicine students as being part of medical students; (iii) provided insufficient information to calculate aggregate prevalence; (iv) published articles were inaccessible for full review, and (vi) interventional studies, case reports, case series and commentaries. We also did not consider studies with outcome measures of test anxiety and death anxiety. This was because we considered these measures of anxiety to be situation-specific and issue-specific which did not measure the general state of anxiety which we were interested in. We also excluded studies which measured anxiety during or after a major event such as a war [14], natural disaster [15] or a disease epidemic [16,17] due to the confounder effect of such events.

### 2.5. Selection of Articles

Two of us (T.T.-C.Q. and C.S.-H.H.) carried out the initial selection and vetting of the articles according to the titles and abstracts independently. Irrelevant articles were excluded. We then retrieved the full texts of the remaining articles and appraised the studies based on their study design and evaluated them against the inclusion and exclusion criteria independently. We resolved any differences of opinion on the inclusion of articles in discussions with R.C.-M.H. The selection process, based on the Preferred Reporting Items for Systematic Reviews and Meta-Analyses (PRISMA) guidelines [18], is detailed in Figure 1. The PRISMA checklist for this meta-analysis is shown in Appendix A.

### 2.6. Data Extraction and Study Evaluation

Two of us (T.T.-C.Q. and W.W.-S.T.) further extracted the data reported by the selected articles, and documented the following details in a standardized table: (i) information of publication (last name of first author, year and location of study); (ii) the sample size of the study; (iii) prevalence of anxiety in medical students and in associated sub-groups if available; (iv) instrument used to assess anxiety; (v) mean age of sample size; (vi) proportion of female and single medical students, and (vii) prevalence of anxiety in non-medical students if available. For studies reporting prevalence at different periods in time within the same sample, the estimate at baseline was extracted except where more informative estimates were provided at a later stage. To appraise the quality of the selected studies, we adopted an adapted version of the Newcastle-Ottawa cohort scale for cross-sectional studies [19]. This adapted scale takes into consideration the selection of samples, comparability of subgroups and evaluation of outcome measures. Using this scale, we scored each study independently. We then reconciled scoring differences through discussion. 

### 2.7. Statistical Analysis

We utilized the software Comprehensive Meta-Analysis (CMA) Version 2.0 (Biostat, Inc., Englewood, NJ, USA) to perform the statistical analyses. Before statistical analysis was carried out, we randomized all articles by blinding the title of the articles and the names of the authors and the journal. The selected studies reported the dichotomous variable of anxiety being present or absent in medical students based on the study authors’ defined cut off score for the selected screening instruments. Using the random-effects model, we then calculated the aggregate prevalence of anxiety, the corresponding *p*-value and 95% confidence interval (CI), the Cochran’s Q-statistic and its *p*-value. We used the random-effects model due to the expected heterogeneity across the studies and also because it considers subject-specific effects [20,21]. We used the I^2^ value to assess heterogeneity among studies. When significant heterogeneity was identified, we ran a meta-regression using the random-effects model to study the impact of moderator variables on the prevalence of anxiety in medical students. For studies comparing medical students with non-medical students, we calculated the summary odds ratio (OR) and reported the corresponding *p*-value, 95% CI and Z-value. Statistical tests were two-tailed, and we set the level of significance at 5%. We evaluated publication bias using the Egger’s regression method [22]. If significant publication bias was present, we would perform the classic fail-safe test to determine the number of missing studies required for the *p*-value of publication bias among the observed studies to approximate > 0.05.

### 2.8. Subgroup Analyses

We performed subgroup analyses to study the effects of gender and year of study on the prevalence of anxiety among medical students. We also grouped studies by continent based on the classification provided by United Nations Standard Country or Area Codes for Statistical Use [23]. We differentiated Middle Eastern countries from Asian countries based on traditional definitions. We classified the countries Bahrain, Cyprus, Egypt, Iran, Iraq, Israel, Jordan, Kuwait, Lebanon, Oman, Palestine, Qatar, Saudi Arabia, Syria, Turkey, United Arab Emirates, and Yemen as countries in the Middle East.

## 3. Results

In this meta-analysis, we pooled and analyzed data from sixty-nine studies comprising 40,438 medical students. Eighteen studies from Asia, twenty-one studies from Middle East, thirteen studies from Europe, ten studies from South America, four studies from North America, two studies from Oceania and one study from Africa were included. The sample size of the studies ranged from 73 to 10,140. The mean age of sample size ranged from 18.02 years to 25.10 years. The proportion of female medical students in each study ranged from 0% to 100%. The screening instruments used to assess for the presence of anxiety in medical students included the Beck Anxiety Inventory-21 item Scale (BAI-21) [24], the Hospital Anxiety and Depression Scale (HADS) [25], the Depression Anxiety Stress Scales-21-item-Anxiety Subscale (DASS-21) [26], and the Generalized Anxiety Disorder 7-item Scale (GAD-7) [27]. Table 1 shows the location, prevalence rate of anxiety and the instruments used to assess anxiety in each of the sixty-nine studies examining anxiety in medical students. The mean age, proportion of female medical students, and proportion of single medical students in each of the sixty-nine studies are described in Appendix A.

### 3.1. Quality of Studies

Using the Newcastle-Ottawa cohort scale for cross-sectional studies [19] to evaluate the quality of studies, all sixty-nine studies scored at least 7 points out of a possible 10 points. This quality appraisal, in addition to the provision of ethical approval for each study, is detailed in a table in Appendix A.

### 3.2. Pooled Prevalence

Analyzing the global prevalence rate of anxiety among medical students using the random-effects model, we found the pooled prevalence based on the sixty-nine studies to be 33.8% (95% CI: 29.2–38.7%; I^2^ = 98.79). Figure 2 illustrates the forest plot of the sixty-nine studies reporting the prevalence of anxiety among medical students. As there is substantial heterogeneity among the studies analyzed (I^2^ = 98.79), we carried out meta-regression against mean age (β = 0.024 *p* = 0.78) and proportion of female students (β = 0.21, *p* = 0.80), but none of these moderators had a statistically significant effect on the prevalence of anxiety.

### 3.3. Subgroup Analyses

Table 2 illustrates the results of the subgroup analyses.

When the prevalence was stratified by gender, we found that female medical students had a higher prevalence of anxiety (38.0%, 95% CI: 27.6–49.5%) than male students (27.6%, 95% CI: 19.3–37.8%), but this difference was not statistically significant (*p* = 0.16). Appendix A shows the prevalence rate of anxiety stratified by gender as reported by twenty-four studies examining anxiety in medical students.

When we compared medical students in pre-clinical years with those in clinical years, medical students in clinical years had a slightly higher prevalence of anxiety at 26.4% (95% CI: 20.6–33.1%) compared to those in pre-clinical years at 26.2% (95% CI: 21.2–31.9%), but the difference in prevalence was not statistically significant (*p* = 0.96). Appendix A shows the prevalence rate of anxiety stratified by pre-clinical or clinical year of study as reported by twenty-five studies.

As the number of studies conducted in South America, North America, Europe, and Oceania was relatively small, we grouped these four continents into a subgroup named ‘Rest of the World’ and ran a subgroup analysis with the subgroups Asia and Middle East. When the prevalence was grouped by continent, the prevalence of anxiety, in descending order, was as follows: Middle East at 42.4% (95% CI: 33.3–52.1%), Asia at 35.2% (95% CI: 26.3–45.3%), Rest of the World at 27.5% (95% CI: 21.5–34.5%). The differences of prevalence rate of anxiety among the different continents were statistically significant (*p* = 0.04).

### 3.4. Comparison with Non-Medical Students

Eight studies compared the prevalence of anxiety between medical students and non-medical students. Appendix A illustrates the comparison of prevalence rate of anxiety between medical students and non-medical students as reported by the eight studies. Odds ratios calculated from these studies are represented in Figure 3. The pooled OR between the two groups was statistically insignificant (OR 0.948, 95% CI: 0.648–1.39; *p* = 0.78).

### 3.5. Publication Bias

We found no significant publication bias in the sixty-nine studies (intercept = 3.05, 95% CI: −1.48 to 7.57; t = 1.34; d.f. = 67, *p* = 0.18).

## 4. Discussion

Our findings suggest a high prevalence of anxiety among medical students. A previous systematic review by Hope et al. [12] found the prevalence of anxiety among medical students outside of North America to range between 7.7% and 65.5%. Similarly, a systematic review of anxiety among medical students in North America described a high prevalence rate as compared to age-matched general population [1]. In this study, we found a pooled prevalence of anxiety ranging from 29.2% to 38.7% among medical students globally. As a comparison, the prevalence of anxiety among the general population was found to be 3% as screened by the DASS-21 [96], 8.2% as screened by the GAD-7 [97], not more than 10% as screened by the BAI-21 [98] and not more than 25% as screened by the HADS-A [99].

It is no surprise that medical students experience a much higher prevalence of anxiety compared to the general population. There are many factors that can explain this. Medical schools preselect for people who tend to be more neurotic and perfectionistic [100], and such personality traits predispose an individual to anxiety [101]. Anxiety can be precipitated in situations such as when self-set lofty goals by these ambitious medical students are not met. Other factors like academic workload [102], consequent sleep deprivation [103], financial burden [103], exposure to deaths of patients [104] and student abuse [105] have also been postulated to be possible reasons for medical students’ high rate of anxiety. Medical schools can help medical students by addressing some of the modifiable factors listed above. For example, frequent reminders about sleep hygiene and its effect on mental health can be sent through e-mails to students. School counsellors can check in with students identified to be financially burdened (e.g., students on tuition loan or financial aid) on a frequent basis to find out if they require additional financial help. Medical schools should also have a robust and anonymized platform for students to give feedback on abusive medical educators and readily provide psychological support to students affected.

When comparing the prevalence of anxiety in medical students among different continents, we found that Middle Eastern and Asian medical students had the highest prevalence of anxiety. We hypothesize that this could be due to the differing views and level of acceptance of people with mental illness in different cultures. For instance, it has been described that people in the Middle East value ‘concealing emotion’. Sharing how one feels with another is uncommon in Middle Eastern cultures, leading to stigmatization towards seeking help from a mental health professional [106]. In Asian cultures, being diagnosed with a mental illness is thought to reflect the patient’s family weakness and is perceived to be shameful [107]. In contrast, it was found that people of Caucasian descent have a lower rate of stigmatization towards mental illness than other socio-cultural groups [108]. Hence, the leaders of medical schools have to take into consideration the unique sociocultural context in developing strategies to tackle anxiety among medical students.

We found that the difference in the prevalence of anxiety between male and female medical students was statistically insignificant, even though the global prevalence of anxiety disorders tends to be higher in females than males [109]. Gender was also not a statistically significant moderator on the prevalence of anxiety among medical students. The traditional notion that ‘girls are worriers and boys are carefree and worry-free’ may not hold true in the context of a group of medical students. Attention and educational strategies on how to tackle anxiety should thus be focused equally on both genders.

Similarly, we found no statistically significant difference in prevalence of anxiety between students in their pre-clinical years and students in their clinical years. A cross-sectional study which compared medical students among different years of study [110] found that students in their last year of study exercised most frequently, slept for the longest duration every night and had the greatest number of close friends, suggesting that medical students may learn to develop better self-care skills and maintain a healthy balance between professional life and personal life through time. However, a longitudinal study found that self-reported stress scores increased as medical students progressed through their year of study [111]. Indeed, the transition to the wards in clinical years as a medical student can be as daunting and stress-inducing as the transition from high school or pre-medical education to a rigorous medical curriculum. Training-related mistreatment and abuse by clinical supervisors was one of the postulated reasons why medical students become more prone to anxiety, stress, and depression in clinical years [112]. The clinical years will also be the first time for many medical students to encounter patient deaths and deal with demanding patients and caregivers. As medical students progress through medical training, they will be entrusted with greater responsibilities and hence face higher level of stress. The promotion of available wellness programs and resources for mental health in medical school should then be ramped up as students become more vulnerable to anxiety.

Comparing medical students with non-medical students, we found no statistically significant difference in the prevalence of anxiety between the two groups. However, only eight studies were included in the comparison analysis and the individual studies gave conflicting results. There was also heterogeneity among the comparison groups, including law, dentistry and economics students. More studies comparing the prevalence of anxiety between medical students and non-medical students are needed to draw meaningful conclusions.

In view of our findings, it is especially important to destigmatize mental illnesses and promote help-seeking behavior among medical students, especially when help-seeking behavior may be wrongly perceived by some students as a form of weakness [113]. There can be dire implications if medical students do not seek help and continue to be burdened by anxiety throughout the rest of their medical careers. In a previous study, students with untreated anxiety faced a decline in academic performance, professionalism and empathy towards patients [1]. By extrapolation, once the medical students graduate to become doctors, they could be less competent in both technical (e.g., procedures) and soft (e.g., communication) skills, may be more prone to errors [13], and thus provide a lower quality of care to patients. In addition to the negative externalities to patients, the personal cost of anxiety to the medical students themselves cannot be understated. A lower quality of life [114], loss of relationships [115] and high rate of comorbidity with depression [116] are often associated with anxiety disorders. In addition, anxiety has been implicated in many other conditions including substance use disorders [117], fibromyalgia [118,119], misophonia [120,121] and irritable bowel syndrome [122,123].

We believe that the effort to destigmatize help-seeking behavior for anxiety, and mental health issues in general, should start with the administrators and leaders of medical schools. There should be a clear and publicized stance by medical schools that having a psychiatric illness will not result in demerit points for them in any way in medical school or affect the competitiveness of their residency or medical licensing board application in the future. However, a 2009 review in the United States of America found that 96% of the state medical licensing board applications reviewed required physician applicants to disclose any personal history of psychiatric illnesses [124] and some of the questions cast physician illnesses ‘in a punitive context’. There have been calls for the disclosure of the history of psychiatric illness to be non-compulsory as this is a barrier to medical students and residents seeking care for their mental health [125,126]. In addition, medical schools can provide confidential access to off-site mental health services for students, as it has been found that medical students desire confidentiality and privacy when seeking medical services [112].

In view of medical students having a high prevalence of anxiety, medical schools can consider organizing structured programs that are shown by research to reduce anxiety [127]. Such programs including life skills training [128] and mindfulness therapy [129] have been validated through studies to reduce anxiety levels in medical students. These group programs can be organized for both students and faculty, doubling up as an opportunity for the faculty to touch base with medical students who may be away from their home institutions on clinical rotations in other hospitals.

More doctors opening up and being transparent about their mental health struggles [130] helps to correct the general public perception that doctors must be mentally strong and allows doctors to be vulnerable. By encouraging faculty who are often senior doctors to share their experiences on how to manage mental health, there will be a positive impact on current and future groups of medical students as they will feel less pressure to be stoic and will, in turn, seek help when they are feeling overwhelmed by stress and anxiety. This is particularly pertinent for medical schools located in regions where the culture does not value sharing emotions with one another and people with mental illnesses are still stigmatized, such as in Middle East and Asia [106].

In our opinion, this study has several strengths. Firstly, this study has minimal publication bias. Secondly, to the best of our knowledge, this is the first meta-analysis to study the global prevalence of anxiety among medical students. Our findings serve to raise awareness that anxiety in medical students is a prevalent and unaddressed issue that should spur initiatives from medical schools globally. However, we acknowledge the following limitations in our study. Firstly, many other factors that could be implicated in predisposing a medical student to anxiety, for example family history and emotional trauma, could not be assessed due to the wide variability of factors examined in the studies. Secondly, the studies included in the analysis used standardized screening questionnaires to determine the presence of anxiety in medical students. However, a full diagnostic interview by a qualified mental health professional would still be a more accurate way to detect the presence of anxiety. Thirdly, a wide variety of screening instruments was used in different studies and these studies used different cut-off points for anxiety, resulting in high heterogeneity among studies. Fourthly, few studies examined specific anxiety disorders such as generalized anxiety disorder in a formal way, and thus there was no data on the specific type of anxiety disorder the students may suffer from. Lastly, the heterogeneity of the study populations, the different locations and the unique socio-cultural environment present in each study location could introduce confounders that could affect the self-reporting of anxiety.

## 5. Conclusions

In conclusion, in this meta-analysis, we found the prevalence of anxiety to be 33.8% among medical students globally, which is substantially higher than the general population. Further research should build on our findings and identify risk factors of anxiety in medical students in their respective socio-cultural contexts so that effective screening strategies can be developed to identify and assist affected medical students. Anxiety has dire implications for both the medical student (a future physician) and patients. Administrators and the leaders of medical schools should take the lead in destigmatizing mental illnesses and promoting help-seeking behaviors when students are stressed and anxious.

## Figures and Tables

**Figure 1 ijerph-16-02735-f001:**
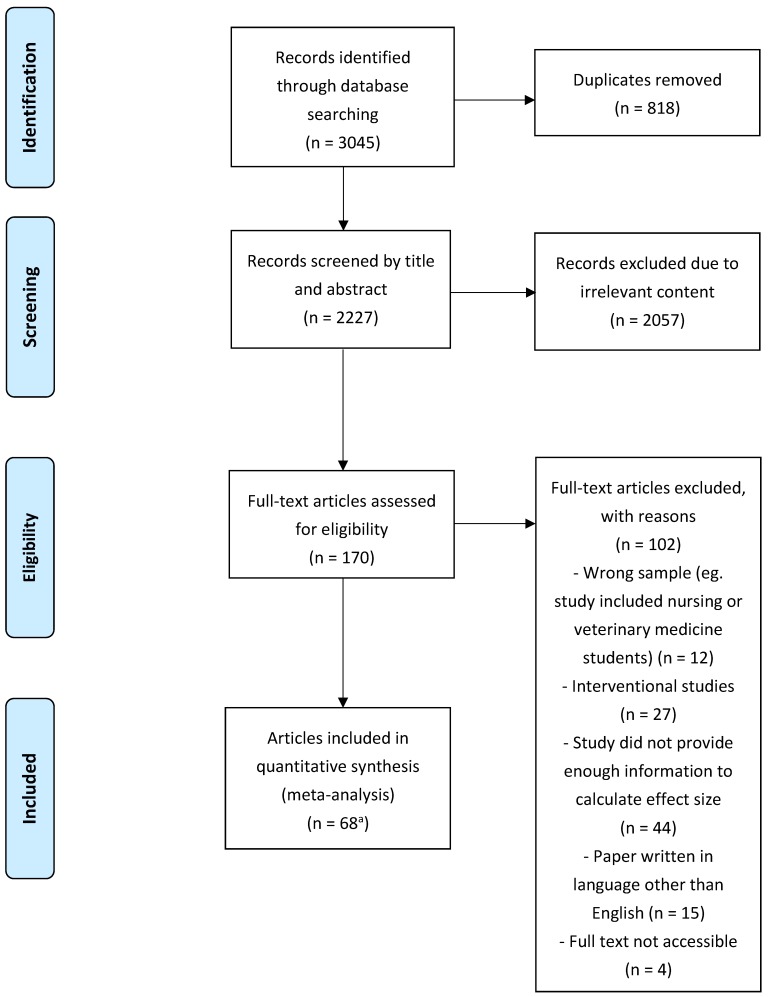
Preferred Reporting Items for Systematic Reviews and Meta-Analyses (PRISMA) flow diagram illustrating the search and selection process of a March 2019 systematic review of the literature on prevalence of anxiety among medical students. ^a^ One article (El-Gilany, 2008) reported two studies.

**Figure 2 ijerph-16-02735-f002:**
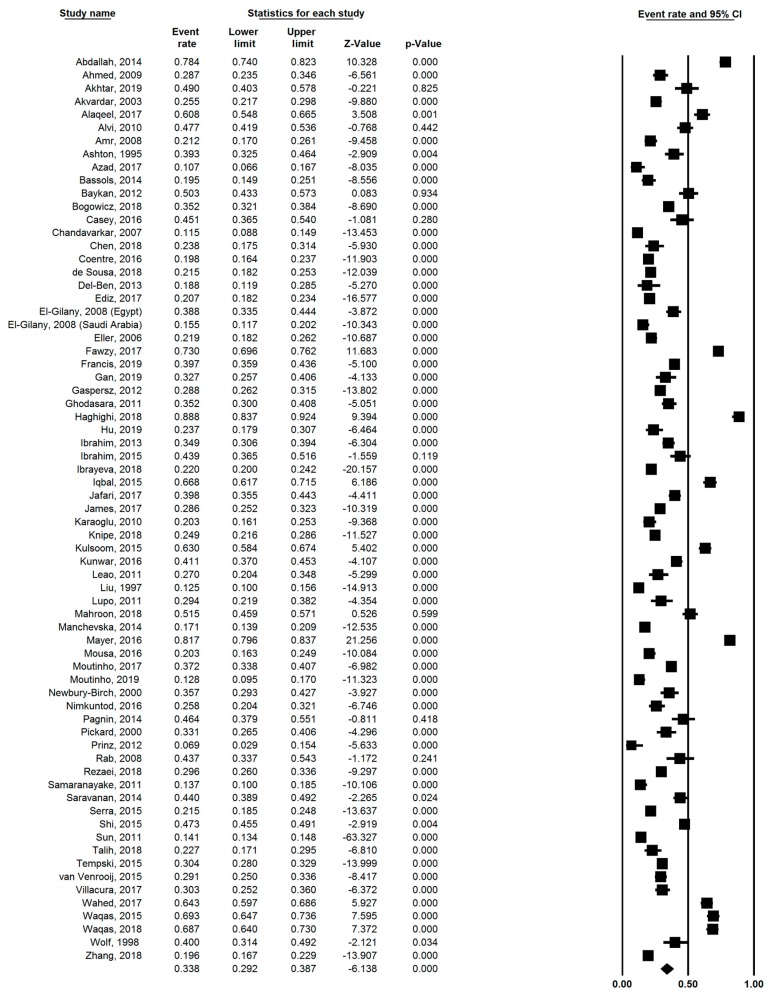
Forest plot of studies (n = 69) examining the prevalence of anxiety among medical students (n = 40,348). 95% CI: = 95% confidence interval.

**Figure 3 ijerph-16-02735-f003:**
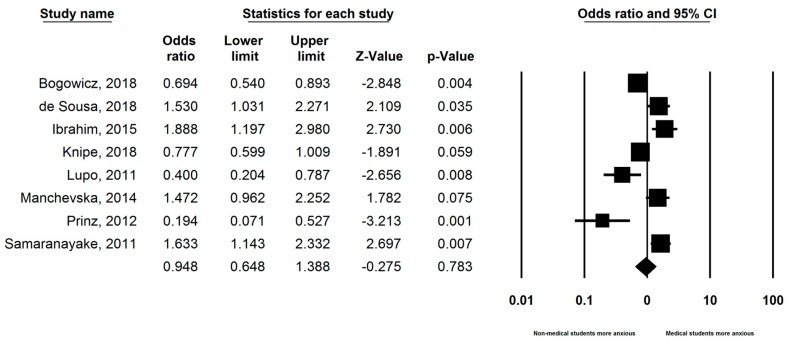
Forest plot of studies (n = 8) comparing the prevalence of anxiety between medical and non-medical students.

**Table 1 ijerph-16-02735-t001:** Location, prevalence rate of anxiety and instrument used to assess anxiety in sixty-nine studies examining anxiety in medical students.

First Author, Year [ref]	Country	Continent	Sample Size	Number of Medical Students with Anxiety	Prevalence of Anxiety, %	Instrument Used ^a^
Abdallah, 2014 [28]	Egypt	Middle East	379	297	78.4	DASS-21-A
Ahmed, 2009 [29]	UAE	Middle East	254	73	28.7	BAI-21
Akhtar, 2019 [30]	Germany	Europe	122	60	49.0	BAI-21
Akvardar, 2003 [31]	Turkey	Middle East	447	114	25.5	HADS-A
Alaqeel, 2017 [32]	Saudi Arabia	Middle East	268	163	60.8	DASS-21-A
Alvi, 2010 [33]	Pakistan	Asia	279	133	47.7	BAI-21
Amr, 2008 [34]	Egypt	Middle East	311	66	21.2	HADS-A
Ashton, 1995 [35]	UK	Europe	186	73	39.3	HADS-A
Azad, 2017 [36]	Pakistan	Asia	150	16	10.7	BAI-21
Bassols, 2014 [37]	Brazil	South America	232	45	19.5	BAI-21
Baykan, 2012 [38]	Turkey	Middle East	193	97	50.3	DASS-21-A
Bogowicz, 2018 [39]	UK	Europe	889	313	35.2	HADS-A
Casey, 2016 [40]	Australia	Oceania	122	55	45.1	DASS-21
Chandavarkar, 2007 [41]	USA	North America	427	49	11.5	STAI-Trait
Chen, 2018 [42]	Taiwan	Asia	143	34	23.8	BAI-21
Coentre, 2016 [43]	Portugal	Europe	456	90	19.8	ZSAS
de Sousa, 2018 [44]	Portugal	Europe	512	121	23.6	HADS-A
Del-Ben, 2013 [45]	Brazil	South America	85	16	18.8	BAI-21
Ediz, 2017 [46]	Turkey	Middle East	928	192	20.7	BAI-21
El-Gilany,2008 [47] (Egypt)	Egypt	Middle East	304	118	38.8	BAI-21
El-Gilany, 2008 [47] (Saudi Arabia)	Saudi Arabia	Middle East	284	44	15.5	BAI-21
Eller, 2006 [48]	Estonia	Europe	413	90	21.9	EST-Q
Fawzy, 2017 [49]	Egypt	Middle East	700	511	73.0	DASS-21-A
Francis, 2019 [50]	Malaysia	Asia	622	247	39.7	HADS-A
Gan, 2019 [51]	Malaysia	Asia	149	49	32.7	HADS-A
Gaspersz, 2012 [52]	Netherlands	Europe	1130	325	28.8	BSI-ANG
Ghodasara, 2011 [53]	USA	North America	301	106	35.2	STAI-State
Haghighi, 2018 [54]	Iran	Middle East	207	184	88.8	STAI-Trait
Hu, 2019 [55]	USA	North America	169	40	23.7	STAI-State
Ibrahim, 2013 [56]	Saudi Arabia	Middle East	450	157	34.9	HADS-A
Ibrahim, 2015 [57]	Egypt	Middle East	164	72	43.9	BAI-21
Ibrayeva, 2018 [58]	Kazakhstan	Asia	1478	325	22.0	GAD-7
Iqbal, 2015 [59]	India	Asia	353	236	66.8	DASS-42-A
Jafari, 2017 [60]	Iran	Middle East	477	190	39.8	DASS-21-A
James, 2017 [61]	Nigeria	Africa	623	178	28.6	HADS-A
Karaoglu, 2010 [62]	Turkey	Middle East	290	59	20.3	BAI-21
Knipe, 2018 [63]	UK	Europe	583	145	24.9	GAD-7
Kulsoom, 2015 [64]	Saudi Arabia	Middle East	442	283	64.0	DASS-21-A
Kunwar, 2016 [65]	Nepal	Asia	538	221	41.1	DASS-21-A
Leao, 2011 [66]	Brazil	South America	144	39	27.0	BAI-21
Liu, 1997 [67]	China	Asia	537	67	12.5	ZSAS
Lupo, 2011 [68]	Israel	Middle East	119	35	29.4	BAI-21
Mahroon, 2018 [69]	Bahrain	Middle East	307	158	51.5	BAI-21
Manchevska, 2014 [70]	Macedonia	Europe	445	76	17.1	BAI-21
Mayer, 2016 [71]	Brazil	South America	1350	1103	81.7	STAI-State
Mousa, 2016 [72]	USA	North America	336	68	20.3	GAD-7
Moutinho, 2017 [73]	Brazil	South America	761	283	37.2	DASS-21-A
Moutinho, 2019 [74]	Brazil	South America	312	40	12.8	DASS-21-A
Newbury-Birch, 2000 [75]	UK	Europe	194	69	35.7	HADS-A
Nimkuntod, 2016 [76]	Thailand	Asia	213	55	25.8	DASS-21-A
Pagnin, 2014 [77]	Brazil	South America	127	59	46.4	BAI-21
Pickard, 2000 [78]	UK	Europe	169	56	33.1	HADS-A
Prinz, 2012 [79]	Germany	Europe	73	5	6.8	HADS-A
Rab, 2008 [80]	Pakistan	Asia	87	38	43.7	BAI-21
Rezaei, 2018 [81]	Iran	Middle East	553	164	29.6	DASS-21-A
Samaranayake, 2011 [82]	New Zealand	Oceania	255	35	13.7	GAD-7
Saravanan, 2014 [83]	Malaysia	Asia	358	158	44.0	DASS-21-A
Serra, 2015 [84]	Brazil	South America	657	141	21.5	BAI-21
Shi, 2015 [85]	China	Asia	2925	1384	47.3	ZSAS
Sun, 2011 [86]	China	Asia	10,140	1430	14.1	BAI-21
Talih, 2018 [87]	Lebanon	Middle East	176	40	22.7	GAD-7
Tempski, 2015 [88]	Brazil	South America	1350	410	30.4	STAI-State
van Venrooij, 2015 [89]	Netherlands	Europe	433	126	29.1	SQ-48-A
Villacura, 2017 [90]	Chile	South America	277	84	30.3	BAI-21
Wahed, 2017 [91]	Egypt	Middle East	442	284	64.3	DASS-21-A
Waqas, 2015 [92]	Pakistan	Asia	409	283	69.3	HADS-A
Waqas, 2018 [93]	Pakistan	Asia	409	281	68.7	HADS-A
Wolf, 1998 [94]	Hong Kong	Asia	114	46	40.0	BAI-21
Zhang, 2018 [95]	China	Asia	616	121	19.6	DASS-21-A

Abbreviations: USA, United States of America; UAE, United Arab Emirates; UK, United Kingdom; HADS-A, Hospital Anxiety and Depression Scale-Anxiety subscale; BAI-21, Beck Anxiety Inventory-21-item Scale; DASS-21-A, Depression Anxiety Stress Scales-21-item-Anxiety subscale; GAD-7, Generalized Anxiety Disorder 7-item Scale; STAI, State-Trait Anxiety Inventory; ZSAS, Zung Self-Rating Anxiety Scale; EST-Q, Emotional State-Questionnaire; BSI-ANG, Brief Symptom Inventory-Anxiety Scale; SQ-48-A, Symptom Questionnaire-48 (Anxiety-Subscale). ^a^ Instrument used to assess presence of anxiety in medical students.

**Table 2 ijerph-16-02735-t002:** Subgroup analyses of prevalence of anxiety in medical students.

Subgroups Comparison	Number of Studies, n	Pooled Prevalence, %	95% Confidence Interval, %	*p*-Value
***Continent***				0.04 *
Asia	18	35.2	26.3–45.3	
Middle East	21	42.4	33.3–52.1	
Rest of the World	30	27.5	21.5–34.5	
***Gender***				0.16
Female	22	38.0	27.6–49.5	
Male	23	27.6	19.3–37.8	
***Year of study***				0.96
Pre-clinical	21	26.2	21.2–31.9	
Clinical	16	26.4	20.6–33.1	

* *p* < 0.05 (two-tailed).

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
