# Peer review of "The Global Prevalence of Anxiety Among Medical Students: A Meta-Analysis"

_ijerph, 2019, doi:10.3390/ijerph16152735_

Round 1
Reviewer 1 Report
The article has potentially valuable information regarding intervention strategies for medical personnel and I found it interesting. Methodology was described in full. However, it is suggested that the authors submit the article to language editors. The article was difficult to read, probably due to translation issues (?). The language used reminds of non-academic language and a thorough editing could improve the message.
Author Response
Response to Reviewer 1’s comments
The article has potentially valuable information regarding intervention strategies for medical personnel and I found it interesting. Methodology was described in full. However, it is suggested that the authors submit the article to language editors. The article was difficult to read, probably due to translation issues (?). The language used reminds of non-academic language and a thorough editing could improve the message.
Thank you for taking time out of your busy schedule to review our manuscript. We appreciate your valuable feedback
We have made stylistic changes and editing throughout the manuscript to improve clarity for readers.
Reviewer 2 Report
In the manuscript entitled “The Global Prevalence of Anxiety among Medical Students: A Meta-Analysis" the authors performed a thorough analysis of the literature in order to determine the global prevalence of anxiety among medical students. I found the paper to be a pleasure to read, and felt as if the articles for the meta-analysis were selected appropriately. Additionally, I appreciate the thorough and interesting discussion.
Minor suggestions include:
1) Consistent tense (anxiety as singular, anxiety disorders as plural)
2) For figure 1, list n's for all groups/boxes, and edit arrows (misaligned on PDF). Also, put footnote in legend, not figure.
3) I believe p 4, lines 216-218 is added in error?
Author Response
Response to Reviewer 2’s comments
In the manuscript entitled “The Global Prevalence of Anxiety among Medical Students: A Meta-Analysis" the authors performed a thorough analysis of the literature in order to determine the global prevalence of anxiety among medical students. I found the paper to be a pleasure to read, and felt as if the articles for the meta-analysis were selected appropriately. Additionally, I appreciate the thorough and interesting discussion.
Thank you for taking time out of your busy schedule to review our manuscript. We appreciate your valuable feedback.
Minor suggestions include:
1) Consistent tense (anxiety as singular, anxiety disorders as plural)
We have corrected this throughout the manuscript.
2) For figure 1, list n's for all groups/boxes, and edit arrows (misaligned on PDF). Also, put footnote in legend, not figure.
We have re-formatted the figure and straightened the arrows. The footnote is now placed after the figure. (pg. 4)
3) I believe p 4, lines 216-218 is added in error?
Yes, thank you for pointing this out. We have deleted the paragraph. (section 3.5)